# Detection of Cancer Stem Cells from Patient Samples

**DOI:** 10.3390/cells14020148

**Published:** 2025-01-20

**Authors:** Sofia Hakala, Anna Hämäläinen, Sanne Sandelin, Nikolaos Giannareas, Elisa Närvä

**Affiliations:** Institute of Biomedicine and FICAN West Cancer Centre Laboratory, University of Turku and Turku University Hospital, FI-20520 Turku, Finland; skhaka@utu.fi (S.H.); anna.e.hamalainen@utu.fi (A.H.); sanne.sandelin@utu.fi (S.S.); nikolaos.giannareas@utu.fi (N.G.)

**Keywords:** cancer stem cell (CSC), diagnostics, immunohistochemistry (IHC), multiplex, cytometry, single-cell RNA sequencing (scRNA-seq), liquid biopsy, immunoassay, medical imaging, spatial transcriptomics

## Abstract

The existence of cancer stem cells (CSCs) in various tumors has become increasingly clear in addition to their prominent role in therapy resistance, metastasis, and recurrence. For early diagnosis, disease progression monitoring, and targeting, there is a high demand for clinical-grade methods for quantitative measurement of CSCs from patient samples. Despite years of active research, standard measurement of CSCs has not yet reached clinical settings, especially in the case of solid tumors. This is because detecting this plastic heterogeneous population of cells is not straightforward. This review summarizes various techniques, highlighting their benefits and limitations in detecting CSCs from patient samples. In addition, methods designed to detect CSCs based on secreted and niche-associated signaling factors are reviewed. Spatial and single-cell methods for analyzing patient tumor tissues and noninvasive techniques such as liquid biopsy and in vivo imaging are discussed. Additionally, methods recently established in laboratories, preclinical studies, and clinical assays are covered. Finally, we discuss the characteristics of an ideal method as we look toward the future.

## 1. Introduction

Cancer stem cells (CSCs) have been studied for the past thirty years. From times of theory and speculation, the existence and importance of CSCs have been recognized in all major cancer types [1,2]. CSCs are a population of therapy-resistant, plastic, self-renewing cells capable of initiating, metastasizing, and forming heterogeneous tumors [3]. Due to these characteristics, CSCs encompass significant clinical value. Therefore, the detection of CSCs from patient samples is becoming increasingly important. It is central not only to have reliable methods to estimate the effectiveness of targeted therapies but also to enable monitoring of disease progression and early diagnosis.

The main strategies that have been used to identify CSCs are based on their tumor initiation capacity (xenotransplantation), self-renewal ability (organoid formation), secretion efficiency (drug efflux), and expression of CSC-specific factors (proteins, nucleotides, and glycovariants). Of these, CSC-specific markers would be most optimal for clinical use. Over the years, a full range of CSC markers across various cancer types have been identified, and the list continues to expand. The most studied markers in clinical samples are CD44, CD133, ALDH, CD24, CD34, and EpCAM, whereas the expression of pluripotent core factors such as NANOG, SOX2, and OCT4 has become more evident in recent years [4,5,6]. In addition, the CSC spatial microenvironment, or “niche”, which is crucial for CSC signaling and function [7,8], provides an alternative way to identify CSCs through various signaling factors and associated cell types.

One major challenge for quantitative CSC detection is the high inter- and intrapatient heterogeneity and lack of exclusive CSC markers. However, single-cell-based multiparameter methods can change these limitations into a successful detection method that could quantitatively detect the full range stemness. This review aims to highlight CSC detection methods (Figure 1) that would be most applicable for clinical use considering the special characteristics of CSCs.

## 2. Spatial Analysis Methods for Tissues

Invasive operation is always performed before analysis of patient tissues. However, surgical removal of tumors is a standard operation, where excessive material is often available. In addition, fine-needle puncture can be used to obtain tissue that can be used for diagnosis or monitoring of disease progression.

### 2.1. Conventional Immunohistochemistry

Chromogenic immunohistochemistry (IHC) is a straightforward, accessible, and affordable standard method used in clinical practice for analyzing tissue samples. Importantly, the stained slides are permanent and can be archived for long periods. The main limitation of standard IHC is the restricted number of markers that can be used (one to two per slide). This limitation can be alleviated to some extent by utilizing consecutive serial sections. Further, the linear dynamic range of the signal intensity observed by brightfield is narrow, resulting in semiquantitative data. In addition, IHC shows interlaboratory variance and requires a trained pathologist for analysis, which can be reduced to some extent by utilizing modern digital analysis techniques [9,10].

Nevertheless, IHC has been the most commonly used technique for analyzing CSCs from patient samples. Despite single-marker data, such as CD44, CD133, or ALDH1, a correlation between survival and CSC expression can be demonstrated based on a meta-study involving over 100 IHC patient studies [1].

### 2.2. Multiplex Techniques

Multiplex techniques increase the number of markers to be used simultaneously and could provide a cost-effective method to detect heterogeneous subpopulations of CSCs from patient tissues. Moreover, multiplexing includes spatial information on CSCs and niche-associated cell types such as immune cells in their natural microenvironment. There are various multiplexing techniques available that are compatible with standard clinical instruments such as multiplex immunohistochemistry (mIHC), multiplex immunofluorescence (mIF) and multiplexed immunohistochemical consecutive staining on a single slide (MICSSS). Of these, mIHC and mIF are based on tyramide signal amplification, enabling the visualization of four to five markers simultaneously. MICSSS increases the number of markers to ten, but it also extends the time required due to the multiple staining rounds, which can also affect tissue integrity and antigenicity [10,11].

mIHC was recognized already in 2010 as a method to identify high-risk patients with breast cancer by identifying putative CSCs [12]. Additionally, mIHC has been used to study CSCs in gastric, cervical, ovarian, non-small lung, and head and neck cancer [13,14,15,16,17] and to expose the stem cell niche [18].

In addition to the conventional multiplex techniques, several higher plex technologies based on mass spectrometry, oligo-barcoded antibodies, and fluorescence are on their way to clinic settings that require special instrumentation and expertise [10,19]. While these technologies significantly increase the number of markers, they also raise costs, optimization time, and analysis complexity. In addition, the tissue area and resolution obtained with these technologies are still limited. However, advanced multiplex technologies may soon provide valuable insights into CSC biology. In the next section, we highlight the possibilities of spatial transcriptomics to capture CSCs.

### 2.3. Spatial Transcriptomics

Spatial transcriptomics covers all the methods that assign transcriptomics data to the original location within a tissue [20,21]. This can be obtained through various approaches such as direct sequencing of microdissected areas and methods that involve in situ hybridization [22,23], in situ sequencing [24], and in situ capturing [25] followed by computational reconstruction of spatial data. Spatial transcriptomics is especially suitable for studying CSC heterogeneity and revealing the spatial organization of niches and other cell types. Importantly, through unbiased sequencing, new characteristics of CSCs can be identified. Limitations of this technology include the inability to perfectly resolve single cells, computational artifacts, the requirement for specific instrumentation, demanding optimization, and cost [20,25].

The spatial transcriptomics platform Visium 10X genomics has been used to reveal radial glial stem-like cells in the neuron-rich invasive niche of glioblastoma [26], the co-location of CSCs and SPP1+ macrophage in the hypoxic region that determines the poor prognosis of hepatocellular carcinoma [27], and the uniform location of CSCs in HPV-negative oral squamous cell carcinoma [28] and to identify and characterize Wilms Tumor CSCs in kidney cancer [29]. Additionally, microdissection and sequencing have revealed the existence of distinct CSC-like populations in triple-negative breast cancer MDA-MB-231 xenografts, also confirmed in single-cell sequencing patient data [30]. Therefore, multiomic approaches can be used to support the initial findings. For example, a spatial multiomic approach was used to reveal a subpopulation of fibroblasts associated with cancer stemness in human hepatocellular carcinoma [31].

## 3. Single-Cell Suspension Analysis Methods

Isolating suspension cells from body fluids, especially blood, is a relatively noninvasive operation for a patient. Through careful tumor dissociation, single cells can also be obtained from solid tumors via invasive procedures like tumor biopsies and surgeries. This section discusses the following primary techniques for single-cell analysis: flow cytometry, mass cytometry, and single-cell RNA sequencing (scRNA-seq). Other technologies applicable after liquid biopsy are reviewed later.

### 3.1. Flow Cytometry

Flow cytometry is a powerful analytical technique widely used in CSC research and clinical analysis, especially in the cases of peripheral blood, bone marrow aspirate, and cerebrospinal fluid. Flow cytometry enables the simultaneous analysis of multiple cellular features by measuring specific surface markers, intracellular proteins, and metabolic functions [32,33,34,35,36]. Flow cytometry is highly sensitive [37,38], allowing researchers to identify and isolate rare CSC populations from blood or tumor tissues. Moreover, the technique is versatile, as it can adapt to various markers for different patient tumor types [39]. Despite its strengths, flow cytometry has limitations. One limitation is the dependency on prior knowledge of marker expression. This reliance can hinder the discovery of unexpected CSC patterns, necessitating additional studies that are time-intensive and constrained by limited sample availability. Further, the method can analyze only single viable cells, and artifacts generated by tissue dissociation processing remain a challenge [40], which can be reduced by using dissociation reagents that help preserve the integrity of cell surface markers [41,42]. One drawback is its restriction to analyzing only one to twelve markers per panel due to overlapping fluorochrome spectra, posing challenges for small tumor biopsies [43]. However, spectral flow cytometry, an invasive technique used in immune cell populations, overcomes this by capturing high-resolution spectral data from cells, recording full emission spectra across all wavelengths instead of a single emission peak [44]. This enables the use of 30–50 markers per panel, conserving limited sample material and allowing more comprehensive analyses than traditional cytometry [45,46,47,48,49]. However, normalization is a key challenge in spectral flow cytometry [50,51]. Additionally, instrumental issues, such as photon collection inefficiencies and inconsistent calibration over multiple days, can lead to technical noise and errors like spreading artifacts obscuring biological signals [52,53,54].

The classical identification of CSCs through flow cytometry relies on detecting a side population, a small subset of cells that actively exclude DNA-binding dyes such as Hoechst 33342 due to heightened ABC transporter activity [35,55]. ABC transporters are integral to the functionality of CSCs and chemoresistance, particularly in their capacity to expel drugs and other compounds [56,57,58]. The side population approach has been employed in various cancer types, including cancer cell lines of breast, glioma, colorectal, and B-cell chronic lymphocytic leukemia patients [59,60,61,62]. Nonetheless, side population analysis poses challenges. Not all CSCs are identified with this analysis, and results are influenced by variability in staining protocols, dye concentrations, and gating techniques [63,64]. In addition, autofluorescence caused by riboflavin accumulation in membrane-bound cytoplasmic structures bearing ATP-dependent ABCG2 transporters can be used to isolate CSCs by flow cytometry [65]. Interestingly, these autofluorescent cells isolated from tumors are distinct from the classical side population.

The classical study by Lapidot et al. (1994) used flow cytometry to show for the first time the existence of CSCs by isolating CD34-positive and CD38-negative cells from patients with human acute myeloid leukemia [66]. Flow cytometry analysis is currently an important and indispensable tool in the diagnosis of mature B-cell lymphomas and leukemias [67]. Recently, a high-dimensional spectral flow cytometry study with 20 antibodies was reported to enable the identification of rare subpopulations that are significant to hematopoietic stem cell hierarchies [47].

Furthermore, flow cytometry is an important tool for isolating rare cell populations from tumor masses and identifying the CSCs using markers such as CD34, CD133, and CD44, along with tumor-specific markers [68,69,70,71,72]. For the characterization of CSCs, marker CD133 has been used for colorectal and brain tumors [73,74,75] whereas a combination of CD44 and CD24 has been used as a standard tool in the case of breast cancers [76,77]. Similarly, for prostate cancer, the CSCs are identified using markers CD133, CD44, and integrin α2β1 [78,79].

### 3.2. Mass Cytometry

Mass cytometry, also known as cytometry by time of flight (CyTOF), is a cutting-edge technology designed for single-cell profiling. The technology has been used mainly for immune cell profiling and is in use in over 200 clinical studies. By leveraging antibodies conjugated to heavy metal isotopes, mass cytometry enables the detection of 40–60 proteins and their post-translational modifications within individual cells. These labeled cells are ionized in a mass spectrometer, where chemical bonds are broken, and metal ions are separated based on their mass-to-charge ratio. This approach permits the simultaneous analysis of numerous markers in a single sample, surpassing the capabilities of traditional flow cytometry [80,81,82]. Therefore, the technique could be invaluable for exploring stem cell heterogeneity.

Mass cytometry offers several advantages over traditional approaches. Unlike fluorescent-based methods, it avoids spectral overlap, allowing the simultaneous quantification of multiple proteins at the single-cell level [83]. Furthermore, its capacity to analyze many parameters in a single sample reduces the number of cells required per experiment, making it suitable for small or precious samples [80]. The ability to characterize spatial marker distribution with imaging mass cytometry adds a valuable dimension to studying tissue organization [84,85]. As with all techniques, mass cytometry has limitations. The destruction of cells during analysis precludes recovery for downstream applications [80]. The flow rate is slower than that of flow cytometry, processing approximately 500 cells per second versus several thousand by flow cytometers. Mass cytometry is also less sensitive in detecting low-abundance molecules, and the reliance on heavy metal isotopes restricts the ability to fully utilize the theoretical capacity of up to 135 channels [80,81]. Additionally, the instruments and reagents are costly, making them less accessible to clinic settings than other single-cell technologies.

The study on glioblastoma stem cells from patient samples underscores the critical role of mass cytometry in understanding the complexity and heterogeneity of CSCs. Using mass cytometry, researchers identified 15 distinct CSC subpopulations based on markers such as CD15, CD44, CD133, and α6 integrin [86]. Notably, the subpopulation expressing all four markers exhibited the highest self-renewal capacity and in vivo tumorigenicity, highlighting its potential therapeutic relevance. Additionally, the study revealed differences in stemness affecting signaling pathways, including MEK/ERK, Wnt, and Akt. Another study of muscle-invasive bladder cancer identified a unique subset of CSCs (CD274+, ALDH+) in tumor tissue using mass cytometry [87]. This contributed to the understanding of the tumor microenvironment and the identification of key cell populations relevant to prognosis and immunotherapy response in this cancer type. Overall, these findings demonstrate that mass cytometry offers a more detailed and comprehensive view of CSCs than traditional methods, allowing the identification of subpopulations that might otherwise go unnoticed.

### 3.3. Single-Cell RNA Sequencing

ScRNA-seq is a transformative technology enabling detailed examination of gene expression at the level of individual cells. Unlike bulk genomic methods that average gene expression across cell populations, scRNA-seq captures cellular heterogeneity and provides insights into processes like stem cell differentiation and drug responses in neighboring cells [88,89]. First introduced in 2009, this technology has become more accessible and widely used, allowing applications such as identifying rare cell populations and revealing gene splicing patterns, co-regulated gene modules, and single-allele expression [90,91,92]. The workflow typically includes cell isolation, messenger RNA (mRNA) capture, sequencing, and bioinformatics analysis [93]. Different methods like Drop-seq, SCRB-seq, and Smart-seq2 offer varying levels of sensitivity, sequencing depth, and cost, enabling customization for specific research needs [94,95,96,97,98]. However, scRNA-seq has its limitations and challenges considering clinical use. To maintain high RNA quality, tissue samples must be dissociated and processed immediately after collection. This poses logistical challenges, particularly in clinical environments where staff may not have specialized training or access to the necessary equipment. The data analysis requires significant bioinformatics expertise and computational resources, which may limit accessibility [99,100,101]. Furthermore, the limited number of cells analyzed can reduce the statistical power needed to detect rare populations [102]. Transcription-level data does not always correlate with functional protein levels in the samples [103], and it also lacks spatial information about cell organization within tissues—an essential aspect for addressing certain biological questions.

Nevertheless, scRNA-seq has been instrumental in advancing our understanding of CSCs [104]. Patel et al. (2014) uncovered the extensive cellular diversity and continuous stem cell-related expression patterns in glioblastoma tumors using scRNA-seq [105]. In bladder cancer, Yang et al. (2017) used scRNA-seq to successfully identify CSCs with clonal homogeneity and mutations that enhance their self-renewal capabilities [106]. Similarly, studies on triple-negative breast cancers have demonstrated significant inter- and intratumoral heterogeneity [107,108].

Pan et al. (2020) applied scRNA-seq and pseudotemporal trajectory analysis to study CSCs in collecting duct renal cell carcinoma, identifying CSC markers (e.g., BIRC5 and CENPF) linked to poor prognosis [109]. Similarly, Fendler et al. (2020) used scRNA-seq in clear cell renal cell carcinoma, revealing Wnt and Notch activation in CSCs and evaluating inhibitors in sphere cultures, organoids, and patient-derived xenograft models [110]. Both studies highlight CSC heterogeneity and therapeutic strategies to overcome drug resistance. Moreover, a recent study utilized scRNA-seq on bone marrow samples from 41 patients with primary refractory multiple myeloma to investigate resistance mechanisms [111]. The study provided insights into early unresponsiveness or progression during first-line treatment, guiding strategies to resensitize tumors to therapies. It also highlights the potential of RNA sequencing to improve resistance prediction and therapeutic personalization in clinical practice.

## 4. Analysis Methods for Biofluids

Liquid biopsy can be defined as the sampling method for any biofluid of the body. In addition to blood, serum, plasma, urine, saliva, pleural effusion, and ascites fluid can be classified as liquid biopsy samples. All the components that tumors secrete to blood are together referred to as the tumor circulome [112]. It includes, for example, circulating tumor cells (CTCs), circulating tumor DNA (ctDNA) and RNA (ctRNA), tumor-derived extracellular vesicles (EVs), and protein markers.

Liquid biopsy has gained popularity over tissue biopsy because it offers semi- to non-invasive, cost-effective, and easy sample collection from patients’ biofluids, with a lower risk of complications [113]. Due to its repetitive nature, liquid biopsy results can be even more reliable than traditional tissue biopsy results and provide a more comprehensive representation of spatial and temporal tumor heterogeneity [113,114,115]. The drawbacks of liquid biopsy are its lower sensitivity and specificity than those of tissue biopsy [114]. Concerns have been raised about the detectability of rare cells and tumor circulome biomarkers in comparison to other cell types and biomolecules present in the bloodstream [115,116]. The limitations of liquid biopsies in cancer diagnostics have been reviewed and listed by De Rubis et al. (2019) [113]. In this chapter, we focus on the CSC-related circulome in the blood as well as in other biofluids.

### 4.1. Detection of Circulating CSCs from a Blood Sample

CTCs are cells that have escaped into blood flow from the bulk tumor. Research has shown that certain CTCs exhibit characteristics similar to those of CSCs. These features are found to be more common in CTCs than in primary tumors [117,118,119,120,121,122]. As CTCs represent only a small percentage of the cells circulating in a patient’s blood, and circulating CSCs even smaller, CTC enrichment techniques are required for the isolation and analysis of CSCs.

The enrichment methods can be divided into positive enrichment when choosing CTCs or negative enrichment when deleting all the other cell types from a sample [123]. For instance, the negative selection of the other circulome cells is one possible way to enrich the circulating CSCs from patients’ blood [124,125]. A simple way to exclude red blood cells is to use a red blood cell lysis buffer [126,127]. In addition, the enrichment techniques and detection methods can be divided into label-dependent and label-free methods. Label-free methods can utilize size, deformability, density, or electric charge for the enrichment of the cells. Label-dependent methods mostly rely on the detection of CSC markers via antibodies by immunocapture. Many studies are combining different label-free and label-dependent enrichment and isolation methods to ensure the best result.

#### 4.1.1. Label-Dependent Enrichment Methods for Circulating CSCs

CTCs have been traditionally detected by epithelial cell adhesion molecule EpCAM expression. The first FDA-approved method for detecting CTCs called CellSearch utilizes antibody-coated magnetic beads to detect the expression of EpCAM, CD45, and cytokeratins 8, 18, and 19 [128]. It is used in monitoring the state of patients with metastatic breast, colorectal, or prostate cancer [129,130,131,132]. The CellSearch technique can be combined with epithelial-to-mesenchymal transition (EMT) markers or CSC markers to detect circulating CSCs [133,134].

As EpCAM is the most common CTC marker it is used in many circulating CSC studies in combination with known CSC-associated markers. Koren et al. (2016) studied the non-small-cell lung cancer CTCs using density gradient centrifugation followed by EpCAM-labeled magnetic beads to enrich EpCAM-positive cells [135]. The gene expression profiles analyzed by quantitative reverse transcription polymerase chain reaction (qRT-PCR) showed overexpression of CSC marker ALDH1A1 in all the patients. Similarly, Tian et al. (2018) used magnetic beads labeled with EpCAM antibody to capture the CTCs from the blood of non-small cell lung cancer patients [136]. This study defined ALDH1 as driving the stemness of the CTCs and its expression as a prognostic marker for non-small cell lung cancer patients.

Hassan et al. (2022) studied the liver CSCs by analyzing EpCAM and CD133 expression by flow cytometry and stemness-associated microRNA (miRNA) expression levels (miR-1290 and miR-1825) by real-time PCR from the patient’s blood [137]. They concluded that EpCAM and CD133, as well as examined miRNAs, could serve as diagnostic and prognostic markers for liver cirrhosis or hepatocellular carcinoma. Varillas et al. (2019) studied CTCs and CSCs in pancreatic ductal adenocarcinoma with a geometrically enhanced mixing microfluidic chip containing antibodies against EpCAM and CD133 [138]. Additionally, they used label-free density gradient centrifugation to enrich the cells. A total of 78% of their samples were positive for the CSC marker CD133. They showed that although circulating CSCs are rare compared to other cells in a patient’s blood, they can still be isolated and identified.

Immuno-based methods can also be used for the negative enrichment of CSCs [139,140]. Mihalcioiu et al. (2023) utilized blood apheresis to collect CTCs from selected breast cancer patients to find CTCs with metastatic abilities [141]. Cells remaining after CD45-negative selection expressed markers such as CK8, EpCAM, ALDH1, CD44, and CXCR4.

#### 4.1.2. Label-Free Enrichment Methods for Circulating CSCs

The most used method for label-free enrichment of CTCs and circulating CSCs from blood samples is density gradient centrifugation as it is a simple and cost-effective standard method for blood samples [127,142,143,144]. Some of the label-free enrichment methods are partly developed to overcome the limitations of EpCAM-dependent selection as some of the circulating CSCs could fully or partially undergo the EMT and lose the expression of EpCAM in the process [140,141,145,146].

Microfluidic chips can be designed as label-free. Hyun et al. (2016) studied CTCs in metastatic breast cancer patients’ blood using a parallel multi-orifice flow fractionation (p-MOFF) chip that characterized the cells by size, shape, and density [145]. By further analysis, they identified patients’ CTCs positive for CSC marker ALDH1A1. Notably, only 4 of the 24 patients had solely EpCAM-positive CTCs, which underlines the weakness of EpCAM in CSC detection. Lin et al. (2017) developed a microfluidic labyrinth that uses size-based separation [147]. They were able to separate subpopulations of CTCs and found CSC-like cells by gene expression analysis. The labyrinth technology was later used by Wan et al. (2019) to analyze CTCs from hepatocellular carcinoma [148]. As they performed immunostaining, they were able to detect CD44-positive populations in all stages of the cancer. Also, the FDA-cleared Parsortix PC1 system is a label-free microfluidic-based technique for cell categorization by size and deformability [149,150]. This technique was later used to isolate the CTCs of non-small cell lung cancer and then analyze the gene expression profiles with qPCR. The results showed that the CSC markers NANOG and PROM1 were associated with a poor prognosis [118].

Another label-free cell enrichment tool utilizes dielectrophoresis and field-flow to detect differences in polarization and categorize cells from blood samples [151]. This tool has been used to detect CTCs that have passed through the EMT or gained the CSC phenotype in early-stage breast cancer [152]. Of 47 patients, 55–74% had epithelial CTCs (CK+, EpCAM+, or E-cadherin+), 57–72% had EMT-CTCs (vimentin+ or β-catenin+), and 9–22% had CSC-CTCs (CD44+ and CD24^low^) based on immunofluorescence staining.

### 4.2. Detection of CSCs from Other Biofluids

Depending on the type of cancer, CSCs can also be detected and isolated from other biofluids. In particular, ovarian cancer research has been using the ascites fluid to isolate the CSCs [153,154,155,156,157]. In these studies, ascites CSCs have been isolated utilizing, e.g., centrifugation and negative enrichment techniques followed by enzyme-linked immunosorbent assay (ELISA), qRT-PCR, or flow cytometry analysis. In addition, isolated CSCs have also been studied by spheroid and adherent cell cultures in vitro [154,155,157]. Kuroda et al. (2013) isolated CSC-like cells from ascites using anti-CD326 (EpCAM) microbeads and then measured the ALDH1 activity of the cells [158]. High ALDH1 expression correlated with a poorer prognosis in serous and clear cell adenocarcinoma samples. Studies like these, one after another, have shown the correlation between CSCs and the poor prognosis of ovarian cancers.

In the case of, e.g., breast and lung cancers, malignant pleural effusion can emerge. This fluid has also been shown to contain CSCs, which have been studied directly with IHC, immunofluorescence, RT-PCR, flow cytometry, and genomic DNA analysis [159,160,161,162]. In addition, floating tumor cells with CSC features have been found in cerebrospinal fluid collected with lumbar puncture from breast cancer patients with leptomeningeal metastasis [163]. In this study, CSCs were enriched through centrifugation and subsequently analyzed using flow cytometry. The analysis revealed overexpression of the markers Syndecan-1 (CD138), MUC-1 (CD227), CD45, CD34, CD24, CD44, and CD133.

Biomarkers can also be found in saliva. In some cancer types, especially in oral cancers, tumor-specific DNA, RNA, or proteins are often found in saliva samples [164]. Kamarajan et al. (2017) analyzed saliva together with tissue and plasma samples to study head and neck squamous cell carcinoma CSC metabolomics by UPLC-MS/MS (ultra-performance liquid chromatography–mass spectrometry) [165]. They showed that glutaminolysis plays a role in regulating ALDH levels in these cells and induces stemness.

### 4.3. Detection Methods for CSC-Derived Extracellular Vesicles

EVs released by cells into the extracellular space can contain various surface markers and biomolecules of the cell of origin, making them valuable for cancer diagnosis. CSCs are proposed to have elevated secretion of EVs compared to cancer cells [166,167]. Both cancer cell- and CSC-derived EVs have been shown to promote cancer progression and metastasis [168,169,170,171]. Additionally, CSC-derived EVs have been shown to carry CSC-related RNAs that can cause stemness in surrounding cells [169,172,173,174,175]. The role of CSC-derived EVs in cancer has been extensively reviewed by Naghibi et al. (2023) and Su et al. (2021) [176,177].

By detecting CSC-derived EVs from biofluids, one can predict CSCs’ existence. The most obvious way to detect CSC-derived EVs is to use known EV and CSC markers together. In a recent study, ovarian cancer patient-derived EVs from urine and ascites fluid before and after chemotherapy were isolated and characterized using ultracentrifugation, CD9-coated magnetic beads, and flow cytometry using markers CD81, CD117, and EpCAM [178]. Also, CSC-derived exosomes have been isolated from melanoma patients’ serum and ascites fluid from pancreatic cancer patients with ultracentrifugation and an exosome isolation kit [179,180].

CSC-derived EVs have also been detected using label-free methods. Surface-enhanced Raman scattering (SERS) is a technique that significantly enhances Raman scattering signals [181]. SERS provides a highly sensitive, label-free method that requires minimal sample preparation and allows sample recovery [182]. Haldavnekar et al. (2022) used SERS to detect and trap CSC-derived EVs from cancer patients’ blood by self-functionalized 3D networks of nanosensors [183]. Interestingly, the EVs derived from CSCs differed from those derived from the bulk tumor cancer cells. The same research group later used this technique to detect CSC-derived EVs from glioblastoma and was able to find differences in molecular signatures between non-cancerous EVs and CSC-derived EVs [184]. Remarkably, an extremely small sample volume, 5 µL, was used for the analysis. Although this technique needs more investigation, it could be used on a patient’s plasma or blood to support other diagnostic tools.

### 4.4. Detection Methods for Secreted CSC-Specific Proteins, RNA, and DNA

Detection of circulating secreted proteins has been the most common way to diagnose and monitor cancer. Many studies have used CSC-associated proteins as markers for CSC existence or cancer progression [185,186,187,188,189]. Protein biomarkers can be detected, for instance, by ELISA, Western blot, or mass spectrometry.

Certain freely floating secreted RNAs in biofluids have also been associated with CSC appearance [172,190,191]. For example, CSC-related miRNAs have been detected in the serum of colorectal cancer patients [192], and CSC-related mRNA expression in the urine of bladder cancer patients has been measured by qRT-PCR [193].

ctDNA analysis, by sequencing and amplification-based techniques, has been used for cancer detection and diagnosis for years. Currently, these new nanobiotechnologies, such as SERS, offer an easier and more cost-effective option. These technologies and their potential use for detecting CSC were reviewed earlier by Sun et al. 2024 [116].

## 5. Analysis Methods for Niche-Associated Factors

The CSC niche is part of the tumor microenvironment, which typically consists of various cell types, including cancer cells, stromal and endothelial cells, cancer-associated fibroblasts, extracellular matrix, signaling molecules, intrinsic factors, blood vessels, immune cells, networks of cytokines and growth factors, and other cellular and acellular components, such as exosomes [194,195,196,197]. Currently, the composition of CSCs, immune cells, and niche characteristics in the tumor microenvironment can be studied using various technologies mentioned in this review, including IHC, mIHC [198], scRNA-seq [199,200], flow cytometry [201], whole-body medical imaging (PET and MRI) [202], imaging mass cytometry [84,203], and multiplex ion beam imaging (MIBI) [204]. In addition to these, CSCs can be detected by chemokines and cytokines, which can be detected by ELISA, Luminex, and chromatography assays, which are discussed next.

### 5.1. Immunoassays—ELISA and Luminex

ELISA is an antibody-antigen-based analytical method for qualitative and quantitative analyses. The assays are performed in a plate format, where the blood, plasma, or serum sample antigens are attached to a solid phase. The antigens are allowed to react with specific antibodies, which are then detected by a secondary antibody labeled with an enzyme [205,206]. ELISA is a simple and cost-effective assay with varying specificity, sensitivity, and efficiency. The preparation of specific antibodies can be labor-intensive and costly. In addition, the assay suffers from a high likelihood of false-positive and false-negative results, compromising reliability [205]. Based on ELISA assays, CSCs have been found to express high levels of IL-8, IL-1β, IL-6, TNFα, VEGF, PTGS2, CXCR1, IL-15, and IL-13/IL-13RA2 in the blood and serum samples of cancer patients [207,208,209,210,211,212].

The Luminex assay is a multiplex biometric ELISA-based immunoassay that contains shaded microspheres conjugated to a monoclonal antibody specific to the target protein [206,213]. The Luminex assay can be performed on patient blood, plasma, and serum samples and allows the simultaneous measurement of up to 100 analytes with high sensitivity, a dynamic range, high throughput, and minimal sample requirements [206,214,215]. However, Luminex assays can be complex, and variability may arise from lot numbers, kit components, antibodies, analyte standards, assay performance, instrumentation, data analysis, and calibration [213,214,216]. CSCs in patient blood and plasma samples have been shown to correlate with significantly high levels of the cytokines including IL-1β, IL-2, IL-7, IL-8, IL-15, G-CSF, IFN-γ, TNFα, VEGF, and FGF as measured by Luminex assay [215,217,218,219].

### 5.2. Chromatography

Chromatography is based on the principle that molecules in a mixture applied to a stable phase are separated by the movement of the mobile phase depending on the molecular properties of molecules related to absorption, partitioning, affinity, and differences in molecular weights [220]. The chromatographic method is efficient and rapid but requires detailed knowledge of the substances to be separated [220,221,222].

Chromatographic methods are commonly combined with immunoassays, flow cytometry, and mass cytometry. For instance, molecules are separated by chromatography, such as HPLC (high-performance liquid chromatography), FPLC (fast protein liquid chromatography), and CE (capillary electrophoresis) and detected by mass cytometry [219,221,222]. Chromatographic methods have been used with Luminex to detect elevated levels of the following cytokines in plasma samples from cancer patients: IL-6, IL-17, IFNα, IFNγ, and TNFRII [219].

## 6. Medical Imaging Methods

Non-invasive imaging with positron emission tomography (PET), single-photon emission tomography (SPECT), or magnetic resonance imaging (MRI) has great potential for developing diagnostic methods for CSCs, enabling whole-body imaging revealing CSCs in their natural environment. Non-invasive imaging-based detection would be key, especially for detecting small or widespread metastatic lesions, and provides an opportunity for frequent monitoring of a therapy’s effectiveness. The limited number of CSC markers to be used simultaneously is a clear disadvantage of these techniques, although multi-tracer tumor imaging is possible [202]. Further, the use of tracers requires careful timing, as after intravenous administration, the tracers will slowly move to a specific location. In addition, unspecific signals are generated through the accumulation of tracers in certain organs, such as the liver, bladder, kidneys, and spleen. However, these non-invasive imaging methods could be used to expose patients’ susceptibility to CSC treatments directed against certain receptors. Importantly, antibody/peptide-assisted imaging methods offer the chance for targeted delivery of therapy agents or therapeutic radiation to the tumor.

### 6.1. Magnetic Resonance Imaging

In MRI, CSC-specific antibodies or peptides can be conjugated with superparamagnetic nanoparticles, which can be imaged after intravenous injection, resulting in a resolution of 0.3–2 mm within a tissue. Several options have been tested to image CSCs in vivo in tumor xenograft models. For example, CD133- and EpCAM-positive cells in mice have been successfully imaged in vivo with MRI with antibody-coated nanoparticles [223,224,225]. In addition, hyaluronan-modified magnetic nanoparticles have been developed to image and target CD44-positive cells in mice based on the ability of CD44 receptors to bind hyaluronan [226,227,228]. Additionally, CSCs positive for the fibronectin variant were imaged with variant-specific fibronectin peptide ligand particles [229] and later used for the targeted delivery of chemotherapeutic agents [230]. In addition, increased uptake of ferritin or glutamine by CSCs has been tested to reveal the location of CSCs and as a theranostic strategy with MRI [231,232].

### 6.2. Positron Emission Tomography

For PET or SPECT imaging, CSC-specific antibodies can be conjugated with positron-emitting or gamma-emitting radioactive tracers, reaching a resolution of 0.6–5 mm in vivo depending on the instrument. CD133, CD44, EpCAM, or CXCR4 antibody/peptide coupled with 64Cu, 89Zr, 18F, 68Ga, or 125I tracers have been successfully used to image CSCs in vivo in tumor xenograft models [233,234,235,236,237,238,239,240,241,242,243,244,245,246]. Initial phase I clinical studies using 89Zr-labelled CD44 antibody showed specific tumor uptake with an acceptable safety profile [247,248]. As anticipated, both non-specific and specific uptake was observed in normal tissues. Synthetic peptide 68Ga-PentixaFor, which specifically binds CXCR4, a marker that has been suggested to be a CSC marker, has been used successfully in over 1000 patients in multiple studies to image disease progression in hematologic and solid tumors, as reviewed in Lindenberg et al. 2024 [249].

## 7. Discussion

Based on this extensive review, the detection of CSCs from clinical patient samples is achievable using the various techniques summarized in Figure 2 and Figure 3. Samples for analysis can be obtained directly from tumors, but non-invasive measurements from biofluids or through direct medical imaging can also be used. Importantly, multiple studies have demonstrated that detecting CSCs would be highly relevant for patients. However, some of the traditional analysis methods should be developed further or replaced to achieve clinically relevant results.

CSCs are a plastic population of cells with high heterogeneity. In addition, CSC markers are subject to variability, influenced by genetic or epigenetic changes, environmental conditions, or drug treatments. Therefore, a single miracle marker cannot be found or used, which could lead to drastic underestimation of the quantity of CSCs. The future of CSC isolation and detection is in multiparameter assays. Importantly, the field should be open to new factors to be identified. Instead of classical surface markers, modern technologies might result in the identification of new pan-CSC indicators. Such breakthroughs would significantly advance CSC research and improve the development of assays and optimizations of new targeted cancer therapies.

In addition to the methods discussed in this review, recent technological developments offer intriguing spatial methods that could be especially suitable for CSCs. One of the new potential label-free universal detection techniques for CSCs is their morphological features, which could be analyzed using deep learning tools on patient samples [251]. Before performing this, it is essential to identify CSCs and correlate the results with morphology across a substantial number of patient samples to train the algorithm. To achieve this, several new technologies in addition to the spatial transcriptomic methods discussed earlier have emerged that allow spatial mapping of 50–100 markers in tissue sections and the identification of new factors. Deep visual proteomics (DVP) allows single-cell resolution proteomic profiling from tissue slides [252], and spatial multimodal omics approaches allow simultaneous analysis of transcriptomes with epigenomes or proteomes [253]. An additional benefit of these techniques is the possibility of visualizing niche-associated cells, such as immune cells, that could further support diagnosis.

Recent advancements in single-cell techniques, such as flow cytometry, mass cytometry, scRNA-seq, and higher plex technologies, have significantly enhanced our ability to study CSCs. These multiparameter methods are likely to reveal, in the near future, the true complexity of the inter- and intrapatient heterogeneity of CSCs suggested by initial studies [26,27,28,29,30,31,47,87,88,106,107,108,109,110,111,112]. Importantly, these technologies can result in the identification of new diagnostic and therapeutic targets as multiple markers in new combinations in addition to unbiased sequencing are utilized.

The detection of CSCs, CSC-derived EVs, and CSC biomarkers from biofluids would be highly valuable for cancer diagnosis. Multiple studies have shown the correlation of their existence in patients’ biofluids with metastasis, relapse, and poor prognosis [118,125,136,137,142,144,168,169,170,171,254]. Already existing CTC and EV enrichment detection techniques offer great possibilities for CSC detection from biofluids. However, it is important to note that selecting only EpCAM-positive cells could lead to a loss of a significant proportion of circulating CSCs [140,141,145].

The future of in vivo clinical imaging and targeting of CSCs might be in multimodal PET/MRI devices and the use of super probes [250]. Combining these techniques would allow high spatial resolution and superior visualization of soft tissues via MRI, where outlines of tissues and organs are visible, in addition to extremely high sensitivity and limitless penetration depth via PET, which can also measure biochemical changes. Bimodal PET/MRI super probes could be used to deliver drugs in a targeted fashion, such as through the low pH of tumors or image-guided release. However, many challenges remain to be overcome, such as biocompatibility, pharmacokinetics, targeting efficiency, and toxicity.

The final step in the field is to adopt laboratory assays for standard clinical use. An ideal assay should be non-invasive and produce quantitative data revealing the complete essence of CSC presentation within a patient. In the future, the “stemness” of a patient sample could be as simple and routine as hemoglobin measurement. This goal might be overly optimistic as it would require the identification of a universal panel of stemness indicators resulting in one simple value bypassing the cancer type and inter- and intrapatient heterogeneity and variability. However, the field should strive for an easy clinical readout rather than overly complicated data that can arise from new research tools. Importantly, the results obtained should have prognostic value. Significantly, the data should direct treatment decision-making and point to certain targeted therapies. All this will require collaborative development of CSC-directed therapies along with clinical diagnostic assays. To establish all this, a comprehensive set of assays will be necessary to correlate the full heterogeneity of the CSC burden within the primary tumor, at metastatic sites, and in the blood, along with the secreted factors found in biofluids and observed through medical imaging. To conclude, various assays and large-scale clinical studies will be needed to establish clinically relevant assays.

## Figures and Tables

**Figure 1 cells-14-00148-f001:**
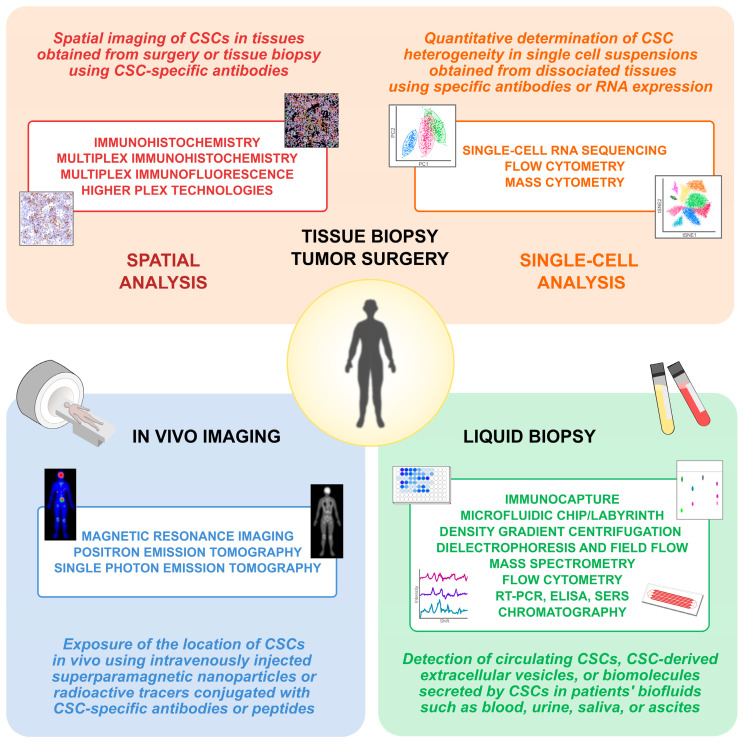
Overview of the methods used to detect CSCs from patient samples. Methods are divided based on sample type into analysis of tumor tissues, liquid biopsy, and in vivo medical imaging.

**Figure 2 cells-14-00148-f002:**
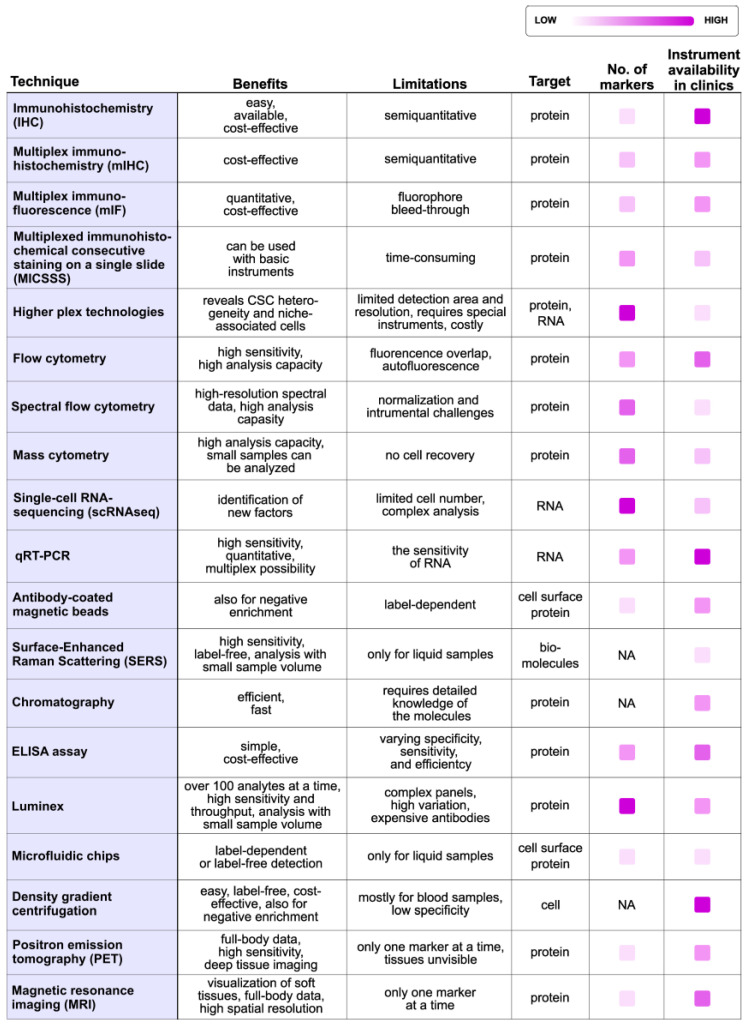
Summary of current techniques for detecting CSCs from patient samples. Immunohistochemistry (IHC) [10], multiplex immunohistochemistry (mIHC) [10], multiplex immunofluorescence (mIF) [10], multiplexed immunohistochemical consecutive staining on a single slide (MICSSS) [10], higher plex technologies [10,25], flow cytometry [33,37,43,45,61], spectral flow cytometry [44], mass cytometry [80,84,86], single-cell RNA sequencing (scRNAseq) [89,91,100,102,111], qRT-PCR [123,192,193], antibody-coated magnetic beads [128,135,136,178], surface-enhanced raman scattering (SERS) [182,183,184], chromatography [220], ELISA assay [205,206], Luminex [206,213,216], microfluidic chips [138,145,147,148,149,150], density gradient centrifu-gation [127,142,143,144] as well as imaging techniques positron emission tomography (PET) [250], and magnetic resonance imaging (MRI) [250] are concluded in this table. NA = Not applicable.

**Figure 3 cells-14-00148-f003:**
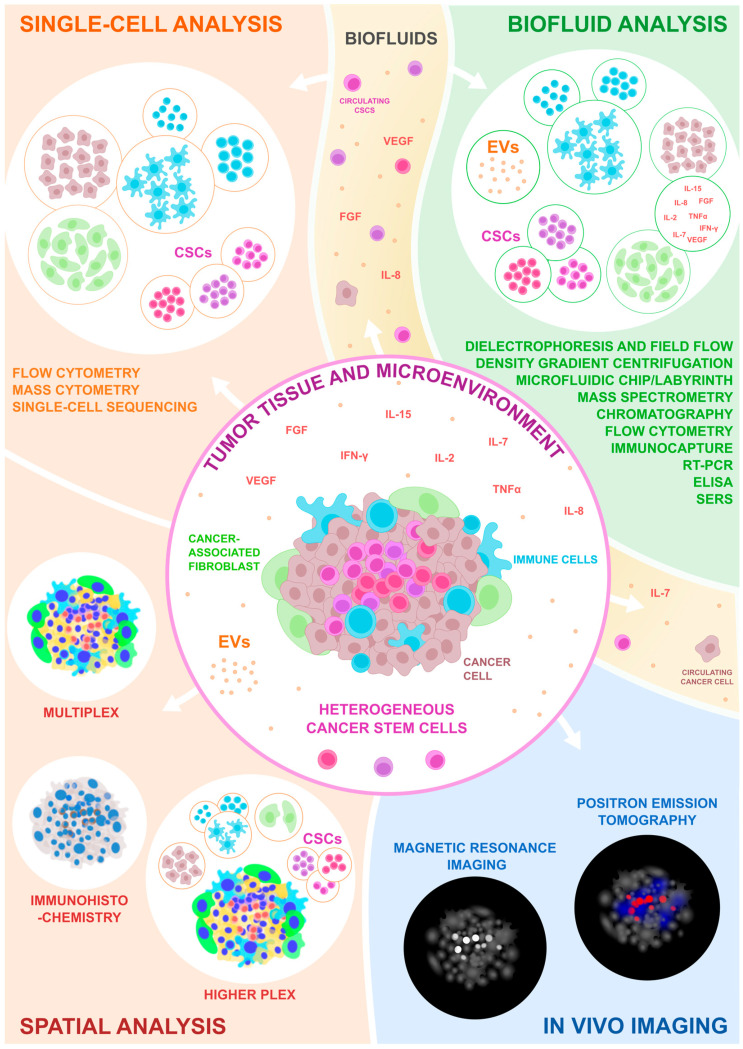
The illustrative detection capacity of currently available techniques to detect CSCs. The techniques are divided into single-cell analysis, biofluid analysis, spatial analysis, and in vivo imaging methods based on the sample type.

## Data Availability

No new data were created or analyzed in this study. Data sharing is not applicable to this article.

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
