# Peer review of "Detection of Cancer Stem Cells from Patient Samples"

_cells, 2025, doi:10.3390/cells14020148_

Round 1

Reviewer 1 Report

Comments and Suggestions for Authors

The manuscript by Hakala at al. provides a comprehensive review of the various techniques available for the detection and analysis of cancer stem cells (CSCs) from patient samples, highlighting their benefits, limitations, and potential future developments, with the goal of establishing clinically relevant assays that can direct treatment decision-making and targeted therapies.

The authors point out that despite active research in this area, standard measurement of CSCs has not yet found clinical application, especially for solid tumors, due to the difficulty of detecting this heterogeneous cell population. Since CSCs play a key role in therapy resistance, metastasis and recurrence, there is a high need for clinical methods to quantify CSCs.

Minor remark

-              What the “stemness” of the blood sample mean?

This needs to be clarified.

In spite authors suggest that in the future, the “stemness” of the blood sample could be as simple and routine as hemoglobin measurement, it seem they are so optimistic.

Comparing the amount of hemoglobin and CSC, it can be said that achieving comparable simplicity is considered doubtful or disputable. Often, a combination of markers and functional assays is used for accurate CSC identification and characterization. According to the reviewer, CSC identification will be based on specific biomarkers, including intracellular and surface markers, and will primarily serve as tools for predicting patient prognosis with respect to specific treatments. Variability in CSC detection will depend on cancer type, CSC origin, tumor stage, and microenvironment, and it is difficult to predict that detection will be universal.

The presented manuscript is well written and contains useful data sources and will be of use to researchers working in the field of CSC.

Reviewer 2 Report

Comments and Suggestions for Authors

Sofia Hakala et al. systematically reviewed various methods for detecting cancer stem cells (CSCs) from patient samples, including spatial analysis of tumor tissues, liquid biopsy, and in vivo imaging techniques. The article highlights the advantages and disadvantages of different detection methods, proposes the characteristics of an ideal detection method, and explores future technological developments. The paper is comprehensive, with extensive references, showcasing the authors' deep understanding of the field. While the overall structure is clear and the data well-reasoned, improvements are needed in language expression, formatting, and scientific details. The following are specific suggestions:

  1. The summary in Table 2 lacks references.
  2. The paper includes too few summary figures; at least 1–2 additional summary figures should be added.
  3. On page 7, line 125, the statement “scRNA-seq has limitations in clinical use” should be supported by relevant data or references.
  4. The discussion of single-cell techniques is too general. It is recommended to include specific literature and detailed analyses, such as case studies of CSC detection in specific cancer types.
  5. In the discussion on page 13, the claim that “combining PET and MRI could offer better visualization” should be further elaborated with specific feasibility and potential challenges.

Reviewer 3 Report

Comments and Suggestions for Authors

This manuscript reviews the strategies and methods for detection of cancer stem cells or cancer stem cell markers from patient samples. First, the authors summarize about spatial methods, single-cell approach, biofluidic method, the detection of niche formation factors, and imaging methods. Overall, this reviewer agrees this review is well written, however, some topics are biased. Addressing these concerns would be more interesting for the readers.

1. The authors summarize about spatial approach. Spatial Transcriptomics technology, such as Visium, Xenium, microdissection with transcriptome analysis, in situ hybridization, and etc., also need to be mentioned and reviewed in the manuscript. For example, PMID: 36823172 revealed glial stem cells in malignant gliomas by spatial transcriptome analysis. On the other hand, spatial transcriptome techniques also has the limitation in the resolution for detection, therefore, some papers performed multi-omics analysis by spatial transcriptomics and single-cell analysis. In this review, the authors mentioned the single-cell analysis, these topics are need be added in the revised manuscript. PMID: 35611554, PMID: 37114795, PMID: 38521868, performed the combination approach of spatial transcriptome and single-cell analysis.

2. In the Table 1, the authors mentioned the number of markers, in details, Flow Cytometry has limited that maximum number is 12. However, Spectral Flow Cytometry is becoming more commonly used now. Spectral FACS enable to detect and sort over 40 colors. This Table need to update with current information or should be delete the column of number of markers.

Round 2

Reviewer 3 Report

Comments and Suggestions for Authors

The authors addressed all my concerns.